# Added Value of *Ascophyllum nodosum* Side Stream Utilization during Seaweed Meal Processing

**DOI:** 10.3390/md20060340

**Published:** 2022-05-24

**Authors:** Anna Þóra Hrólfsdóttir, Sigurjón Arason, Hildur Inga Sveinsdóttir, María Gudjónsdóttir

**Affiliations:** 1Faculty of Food Science and Nutrition, University of Iceland, Aragata 14, 102 Reykjavík, Iceland; sigurjon@matis.is (S.A.); hilduginga@matis.is (H.I.S.); mariagu@hi.is (M.G.); 2Matís Ohf, Food and Biotech R&D, Vínlandsleid 12, 113 Reykjavík, Iceland

**Keywords:** *Ascophyllum nodosum*, brown algae, macroalgae, seasonal variation, proximal composition, trace minerals, monosaccharide composition, bioactive compounds, polyphenols, antioxidant activity

## Abstract

*Ascophyllum nodosum* contains many valuable compounds, including polyphenols, peptides, and carotenoids that have been shown to exhibit biological activities. These compounds are not a priority ingredient in seaweed meal products for the current users. Hence, the aim of the study was to investigate the chemical and bioactive characteristics of *A. nodosum* as affected by seasonal variation and evaluate the potential benefits of alternative processing and the utilization of side streams for product development. The analysis of raw materials, press liquid, and press cake from alternative processing and the commercial seaweed meal at different harvesting periods indicated that the chemical composition is linked to the reproductive state of the algae. Phenolic content and ORAC activity increased following the seaweed’s fertile period, making alternative processing more promising in July and October compared to June. Several valuable ingredients were obtained in the press liquid, including polyphenols, which can be used in the development of new high-value bioactive products. The suggested alternative processing does not have a negative effect on the composition and quality of the current seaweed meal products. Hence, the extraction of valuable ingredients from the fresh biomass during the processing of seaweed meal could be a feasible option to increase the value and sustainability of seaweed processing.

## 1. Introduction

Macroalgae, or seaweed, are plant-like organisms that can be divided into three main categories based on their pigmentation characteristic, i.e., brown seaweed (*Phaeophyceae*), red seaweed (*Rhodophyceae*), and green seaweed (*Chlorophyceae*) [1,2]. At least 221 seaweed species (32 species of green seaweed, 125 species of red seaweed, and 64 species of brown seaweed) are used worldwide in seaweed processing. Generally, seaweed is used for human consumption in three different ways, consumed directly as food (fresh, dried, liquid extracts, canned, salted, or prepared directly as food), as a food supplement, or as a thickening agent (e.g., alginate (from brown algae), agar (from red algae), and carrageenan (from red algae) in a wide range of food products [1,2,3]. As well as direct consumption, seaweed is used as an ingredient in bio-stimulants for agriculture, to enhance nutraceutical properties of plants and fruits for human consumption [4], and to increase plant resilience to abiotic stresses [5]. Seaweed can be both cultivated or wild harvested. The worlds’ production of seaweed in 2005 was 14.7 million tons, where only 1.2 million tons came from wild seaweed. In the year of 2015, the total production doubled to 30.4 million tons, where 29.4 million tons originated from cultivated seaweed, and only 1.1 million tons were obtained from wild resources [1]. Recently, the production has increased even further, and in 2018 the world’s seaweed production reached 32.4 million tons, where 97.1% was obtained from cultivated seaweeds [6].

Seaweed is recognized as a good source of food with high nutritional value. It is known for its high concentrations of carbohydrates, low content of fat, and its richness in polyunsaturated fatty acids, bioactive compounds (including polyphenols, peptides, and carotenoids), vitamins, and minerals [7]. Brown seaweed contains various bioactive compounds that have been proven to have multiple biological activities, such as antioxidant, antiviral, antifungal, and antibacterial properties [8,9]. Seaweed is exposed to a broad range of stressors in its environment, such as fluctuations in temperature, changes in light, desiccation, and osmotic stress. These environmental stress factors can lead to the formation of strong oxidizing agents, such as free radicals. Despite the formation of oxidizing agents, seaweed rarely undergoes major photodynamic damage, which suggests that seaweed cells possess protective mechanisms and compounds [10]. Compounds found in seaweed, such as peptides, low molecular weight sulphated polysaccharides (fucoidan), amino acids, carotenoids, tocopherols, and polyphenols have been reported to have antioxidant activities [9].

Wild brown seaweed, including *Ascophyllum nodosum*, are currently processed into seaweed meal in Iceland and other countries such as Canada and Ireland. In the current production process of seaweed meal, the biomass is mechanically harvested, transported to the factory, chopped, dried, ground, sieved, and packed. The meal is mostly sent out of the country and used in the extraction of the hydrocolloid alginate, used as a fertilizer, or as animal feed. However, to produce 23,000 tons of alginate, approximately 85,000 tons of dried seaweed is needed [11]. The leftover pulp from hydrocolloid production is usually sent to landfills, pumped back to the sea, or occasionally used as soil conditioner [2,12,13], meaning that around 70% of the dried seaweed biomass is not utilized to its full potential. Furthermore, seaweed meal is mainly used as a feed enhancer for animals due to its high content of carbohydrates, vitamins, and minerals (especially iodine) [2] but not due to their bioactive compounds.

Some studies have shown that some bioactive compounds, such as polyphenols, are extractable with water, and extracts retrieved possess antioxidant abilities [14,15]. However, after the extraction of seaweed, many compounds still remain in the leftover biomass, including compounds in the seaweed that are less soluble in water, such as lipids and cellulose [2,16]. Therefore, there might be some opportunities in the valorization of the seaweed meal processing process where an extraction step could possibly be added to part of the process to retrieve valuable compounds from the fresh seaweed biomass to both reduce waste in the seaweed value chain and increase the value of the production. Hence, the aim of the present study was to evaluate the chemical composition and antioxidant activity of *Ascophyllum nodosum* and its produce, as affected by seasonal variation, and to assess the feasibility of developing alternative processes and high-quality side products in a sustainable manner without affecting the buyers’ desired chemical composition of the commercial seaweed meal.

## 2. Results and Discussion

### 2.1. Chemical Composition

The basic chemical composition (moisture, fat, protein, salt, ash, and carbohydrate content) was determined for each harvesting period (June (where the seaweed contained fruit bodies/receptacles), July, October 2020) at four different production stages of *A. nodosum*, i.e., the fresh seaweed, press liquid (PL), press cake (seaweed after extraction), and commercial dried seaweed meal from the production of Thorverk (Table 1). The press liquid and press cake were only measured in July and October due to their antioxidant potentials and their possibilities for product development. The fresh seaweed and seaweed meal were, however, analyzed at all three harvesting periods to evaluate the seasonal variation in *A. nodosum*.

Fresh seaweed generally contains a high moisture content, where the content can reach up to 94% in brown seaweed [17]. In the present study a significant difference was observed in the moisture content of the fresh seaweed between all harvesting periods as well as in the seaweed meal samples. The samples from June contained the highest water content in both fresh seaweed and seaweed meal. The water content might, hence, be related to the reproduction stage of the algae since fruit bodies/receptacles were observed on the seaweed in June, indicating that the seaweed was in its fertile period [18]. As expected, the press liquids contained high moisture contents, or 94.9 ± 0.1 g/100 g sample and 94.8 ± 0.0 g/100 g sample in July and October, respectively. Interestingly, the moisture content observed in the press cake compared well to the fresh seaweed, indicating that the biomass absorbed a significant amount of the added water. The moisture content of the dried seaweed meal was 6.2–9.1 g/100 g sample at all seasons, and thus below the 14–15% threshold as recommended during production for dried feed [19]. When the moisture content of the dried feed/food is lower than 15%, the water activity is lower than 0.6, meaning that there is limited available water for microorganism growth, and the probability of lipid oxidation is low, and hence a stable dried product was obtained [16,20].

Many factors can influence the lipid content and fatty acid composition of seaweed, including temperature, light intensity, and the salinity of the sea. Some studies have also reported seasonal fluctuations where in some species the content is highest in winter, but for others such as fucus species the content has been recorded higher in the summertime [17,21,22,23]. This correlates well with the results from the seaweed meal in the present study, where the content was highest in July and lower in October. The lipid content of seaweed is typically low, where the content for *A. nodosum* has been reported to be in the range between 1.8–4 g/100 g DW [24,25,26]. In the present study, as expected, the total lipid content of the seaweed samples was low in all harvesting periods. The fresh seaweed samples contained 0.2–0.3 g/100 g sample, and the press liquid contained less than 0.01 g lipids/100 g sample at all seasons, indicating that the lipid content in the raw material remained almost completely in the press cake during pressing. Hence any utilization of the press liquid would not affect the lipid amount or composition of the seaweed meal. The low lipid content in the press liquid agrees with the fact that lipids are not water soluble and only dissolve in organic solvents [16]. The low presence of lipids in the press liquid can furthermore be beneficial during the processing and storage of any potential press liquid-derived products, since no or little lipid degradation will happen during storage of the liquid or the dried press liquids.

A protein conversion factor of 5 was used instead of the commonly used 6.25 to calculate the crude protein content. This protein conversion factor was applied, as recommended for seaweed by Angell, et al. [27] due to the high amounts of non-protein nitrogen (NPN) present in seaweed. A significant difference was only observed in the protein content between harvesting times of the seaweed meal, with none in the fresh seaweed, press liquid, or press cake. The protein content was recorded highest in the samples from June, where the seaweed meal contained 6.0 ± 0.3 g protein/100 g sample and the fresh seaweed samples 1.6 ± 0.1 g protein/100 g sample. The results show that some of the proteins goes out with the press liquid during pressing (confirmed by the press liquid protein content of 0.2 ± 0.0 g/100 g sample in both July and October), resulting in a slight lowering in protein content of the press cake. However, the amount of protein extracted with the pressing liquid does not have a significant impact on the protein composition of the press cake when compared to the fresh seaweed samples, indicating that the loss of protein during pressing should not affect the protein yield of the final product substantially. The press liquid, however, contains some protein, which might have some impact on the antioxidant activities of the press liquid extracts since some amino acids and peptides in seaweed have been suggested to have bioactive activities, including antioxidant activities [28,29,30]. Therefore, the protein and peptides might have some effect on antioxidant activities of the press liquid extracts since they have both been reported as potential radical scavengers and ion chelators [28,29,30].

A large part of the seaweed ash content was salt (NaCl), or approximately 17% to 74% of the total ash content. A significant difference was found in salt content between harvesting times in both the fresh seaweed samples and the seaweed meal samples. The salt content of the fresh seaweed and seaweed meal accounted for 29–30% of the total ash content and was recorded to be significantly higher in October (1.7 ± 0.0 g/100 g sample) when compared with June and July (1.3–1.4 g/100 g sample). The salt content of the seaweed meal showed a similar trend to the fresh samples, being lowest in July. Since a large part of the salt content went out with the press liquid, lower salt contents were detected in the press cake compared to the fresh seaweed. The high amounts of salt in the press liquid might affect the use of it in food products, e.g., by intensifying flavors from the extract, especially if used in large quantities. However, if the aim is to use the press liquid as a natural antioxidant in food products, only a small amount would be used, which should not affect the flavor greatly.

The ash content was determined, and the salt free ash content calculated. The salt free ash content of the fresh seaweed samples was significantly differentiated between all three harvesting periods, where the highest content was recorded in July (4.5 ± 0.1 g/100 g sample), followed by October (4.0 ± 0.1 g/100 g sample), and lastly June (3.3 ± 0.1 g/100 g sample). The results, hence, indicate that the ash of the seaweed from July contains proportionally higher quantities of minerals than during the other seasons. Minerals commonly found in brown seaweed include magnesium, sodium, calcium, chlorine, potassium, phosphorus, sulfur, iodine, copper, iron zinc, fluoride, selenium, manganese, molybdenum, nickel, cobalt, and boron [1]. However, in the production of the press liquid, some of the minerals seems to be washed out with the process (0.2–0.3 g/g sample). When the fresh seaweed and press cake were compared, the press cake contained significantly lower amounts of salt free ash, or 28–36% lower values, that would result in lower salt free ash/mineral content of the final seaweed meal as well if the press cake was used in the meal production.

The carbohydrate content of the fresh seaweed samples was significantly differentiated between all harvesting periods, where the highest content was observed in July (22.5 ± 0.2 g/100 g seaweed), followed by October (21.1 ± 0.4 g/100 g seaweed), and lastly June (13.9 ± 0.2 g/100 g seaweed). The same trend was seen in the carbohydrate content of the seaweed meal, where the content was highest in July when compared to the other harvesting periods. The carbohydrate values obtained in June correlated well to values presented earlier [31], that showed a carbohydrate content of *A. nodosum* of 13.1 g/100 g sample. Some carbohydrates seem to go out with the pressing of seaweed, where the content was at an average of 3.5–3.8 g/100 g of press liquid, leaving less content of carbohydrates in the press cake. A significant difference was observed between the carbohydrate content of the fresh seaweed samples from July and October when compared to the press cake samples, which contained 30–35% less carbohydrates, indicating that the extraction process could affect the final carbohydrate content and composition of the final dried product. However, this also shows the potential utilization of the press liquid for the development of products with bioactive characteristics, based on the carbohydrate composition.

### 2.2. Monosaccharide and Uronic Acid Composition

The monosaccharide and uronic acid composition of the four tested production streams were evaluated to predict the final composition in the seaweed meal after extraction (Table 2).

Brown seaweed has been reported to accumulate carbohydrates, and then mannitol, over the summer and autumn months, and the levels decrease in winter [32,33]. Mannitol is a non-hygroscopic sugar alcohol derived from mannose and is naturally present in most species of brown seaweed [9]. It is a straight chain of six carbons with six hydroxyl groups with the chemical formula C6H14O6 [34,35]. The mannitol content in the fresh seaweed samples was significantly lower in June (7.1 ± 0.3% dry weight (dw)) when compared to the other two harvesting periods where similar results were obtained. However, the mannitol content of the seaweed samples was recorded highest in the press liquids (PL), 41.9 ± 3.4% dw and 52.4 ± 2.5% dw in July and October, respectively, leaving the press cake with a low mannitol content.

Glucose units are the main building blocks of the polysaccharide laminarin found in brown seaweed. Laminarin is composed of 20–25 glucose units of (1,3)-β-d-glucan with β-(1,6) branching. They have two main forms, i.e., M (end with mannitol residue) and G (end with glucose residue) chain structures. They are low molecular weight and water-soluble compounds and is one of the main storage polysaccharides in brown seaweed [7,36]. The growth and structure of laminarin depends on both environmental conditions and the species. Specifically, laminarian synthesis of seaweed is directly linked to nitrate and nitrite content in the ocean, which is also connected to the rapid growth phase of the seaweed in the spring. When nitrite/nitrate content is high, the algae grow fast, but when nitrate decreases over the winter a synthesis of laminarin starts and the growth of the seaweed decreases or stops [36,37,38], meaning that laminarin content, and hence the glucose content of brown seaweed, is usually higher in winter as the results from the present study indicate. The glucose seemed to be washed out in some amounts with the press liquid, where the measured content was significantly higher in the press liquid from October (or 32.8 ± 1.2% dw) when compared to the press liquid made in July. However, the press cake seemed to contain similar amounts of glucose after the extraction.

Sulphated fucose (L-fucose 4-sulpfate) is the main building block of fucoidan, which is a sulphated ester polysaccharide. Fucoidan is mainly found in the cell wall of brown seaweed and as an intracellular material. It is both a water-soluble and acid-soluble compound [7,39]. In the present study, the fucose content in the fresh seaweed samples was recorded as significantly higher in October when compared to the other two harvesting periods. Even though the fucose content was higher in October, the press liquid from July contained significantly higher amounts of fucose than the press liquid from October. Some of the fucose, 1.7 ± 0.1% dw and 1.2 ± 0.1% dw, went out with the press liquid in July and October, respectively. However, some of the fucose remained in the press cake. As discussed earlier, fucose is both water soluble and acid soluble. Studies have been performed on different extraction methods for fucoidan that indicate that hot water extraction (with some pre-purification treatment) gives a better yield of fucoidan when compared to acid extraction. The extraction treatments also affect the chemical composition or ratios of the fucoidan [40,41,42]. However, even though water extraction of fucoidan has been more successful than using other solvents, the majority of the fucose content remained in the press cake in the present study. This indicates that factors other than the solvent used in extraction might matter in the case of fucoidan extraction. For example, results from a study conducted by Ferreira, et al. [43] show that the extraction yield increases with a longer extraction time and by using heat in the process. In the present study, no heating was involved, only water at room temperature was used. However, if a higher content of fucoidan was desired in the press liquid, an evaluation of the production process would be required including potential heating steps or an increase in the extraction time.

Xylose/mannose content in the fresh seaweed samples and press liquid was significantly differentiated between harvesting periods, where the content was higher in October in the fresh seaweed samples when compared to the other harvesting periods but higher in July in the press liquid. The content was recorded to be the highest in the press liquids, from 5.8 ± 0.4% dw up to 7.0 ± 1.0% dw. In the other seaweed samples, small amounts of xylose/mannose were recorded of 5% dw or less.

Alginic acid (alginate, algin) is a high molecular and alkali soluble compound. Alginates are linear binary arranged chains made of the uronic acids 1,4-linked β-D-mannuronic (M) and α-L-guluronic acid (G). Alginates are structural components in brown seaweed and are, hence, mainly found in the cell walls but are also an intracellular material. The sequence of M- and G-blocks can vary not only between species but can also be dependent on the harvesting time [44,45]. Alginates are hydrocolloids, and the gel-formation ability of alginates is directly linked to the content of guluronic acids and the length of the G-blocks, where alginates with a higher content and longer G-blocks make gels with a higher strength [16]. In the present study the glucuronic acid content in the fresh seaweed samples was significantly lower in June when compared to the other two harvesting periods. The hexa-mannuronic acid was, however, similar in all harvesting periods. The results, hence, indicate that both the total alginate content and the gel strength in June is lower than the seaweed samples from July and October. The calculated total alginate content of the fresh seaweed samples from June contained the lowest amounts, approximately 11% dw, and the samples from July and October contained approximately 20% dw and 21% dw, respectively. The uronic acid hexa-mannuronic acid was not detected in the press liquids along with only small amounts of Guluronic acid, leaving the majority of the uronic acids in the press cake. Since the press liquid contains such small amounts of uronic acids, the results indicate that the production of the press liquid should not affect the alginate content greatly in the final product.

### 2.3. Trace Elements

One of the main issues in the production of wild seaweed is the possibility of the contamination of heavy metals, such as arsenic and mercury. Iodine is also found in large quantities in seaweed [1,46,47]. Hence, the arsenic (As), cadmium (Cd), mercury (Hg), lead (Pb), and iodine (I) contents were determined in the present study for each harvesting period for the four different production stages. However, as for other measurements of the chemical composition, the press liquid and press cake were only measured in July and October (Table 3).

Arsenic (As) is a metalloid that is found in the environment in both inorganic and organic forms. The organic form is considered to be less toxic than the inorganic one since inorganic arsenic is acknowledged as carcinogenic for humans [48]. In the present study, no significant difference was observed in arsenic content between the harvesting times of the fresh seaweed samples, and the inorganic arsenic content was less than 0.01 in all cases. The total arsenic content of the press liquid was significantly higher in July when compared to the content in October, but was still within the set limits according to the European Commission for dried animal feed. The arsenic content in the press cake was significantly differentiated between the two harvesting periods and contained around 40–50% less arsenic when compared to the fresh seaweed samples. It is known that rinsing food and cooking with high amounts of water, e.g., rice, can reduce the arsenic content of the food component [49], which correlates with the results from the present study, where during the production of the press liquid, the arsenic seemed to be extracted with the water, hence reducing the arsenic content in the press cake. The seaweed meal from June contained significantly lower amounts of arsenic than the other two harvesting periods, 28.5 ± 0.5 mg/kg. Seaweed meal intended for feed should not contain higher amounts of arsenic than 40 mg/kg, where the inorganic arsenic content should not exceed 2 mg/kg [50], but no regulations have been set on the maximum arsenic content of seaweed used as food or supplements for human consumption. Seaweed meal is often used as feed for animals, and, according to the results in the present study, both the inorganic and organic arsenic were below the set limits of 40 mg/kg of organic and 2 mg/kg of inorganic arsenic.

The heavy metal cadmium (Cd) is found in the environment, both naturally occurring and from anthropogenic sources (industrial and agricultural sources). The main source of cadmium exposure for humans, except from smoking, is foodstuffs [51]. In the present study the fresh seaweed samples from October contained significantly lower amounts of cadmium compared to the other two harvesting periods (June and July). The press liquid produced from July-harvested seaweed contained a significantly higher cadmium content, 0.04 ± 0.00 mg/kg, when compared to that from October, which contained less than 0.01 mg/kg, indicating that when a higher Cd content is in the raw material, a higher content will be in the press liquid, leaving less content in the press cake in both cases. However, the seaweed meal from October contained a significantly higher cadmium content than the other two harvesting periods and contained 1.43 ± 0.03 mg/kg of cadmium. The limits of cadmium in seaweed used for animal feed is set to 1 mg/kg [50], meaning that the seaweed meal from both June and July were below that limit but the meal from October exceeded the limits.

Mercury (Hg) is present in the environment, both from anthropogenic and natural sources, in three different chemical forms (metallic mercury, inorganic mercury, and organic mercury) [52]. The mercury content in the present study was recorded to be slightly higher in the seaweed meals independent of season, and in the press cake from July (0.02 mg/kg), than in other samples. However, no significant differences were recorded in the mercury content between the tested processing streams. The set limits of mercury in animal feed with water content below 12% are 10 mg/kg [50] and 0.1 mg/kg for supplements that contain seaweed. Thus, the Hg levels of all seaweed meal samples and press liquid samples were under the limit for mercury to use as both animal feed and as a supplement for human consumption [53].

The environmental contaminant lead (Pb) occurs mainly by anthropogenic sources (including battery manufacturing, mining, and smiting) but also naturally. Humans can be exposed to lead through water, food, soil, dust, and air, but the main source of exposure is through food [54]. A significant difference was observed in the seaweed samples regarding lead content between the harvesting periods, where the seaweed meal harvested in July contained significantly higher amounts, 0.056 ± 0.004 mg/kg, in the meal compared to the other harvesting periods. Slightly higher Pb content was observed in the press cake from October when compared to the press cake obtained in July. However, according to regulations, the lead content of animal feed cannot exceed 0.1 mg/kg (European Commission, 2002). The seaweed meal used in the present study was in no case higher than 0.06 mg/kg, and thus lower than the set limits for lead content at each harvesting periods. The set value for supplements for human consumption is, however, set higher than for animal feed, at 3 mg/kg [53], so all samples were below the limit for supplement production as well.

Iodine (I) is an essential nutrient for the function of the thyroid gland. In the diet, iodine mostly comes from the consumption of seafood [47], and seaweed has been reported to be a very good source of iodine [24,31]. The iodine content in the fresh seaweed samples contained high amounts of iodine, as expected, but the content was differentiated significantly between harvesting periods. The fresh seaweed samples from June contained significantly lower amounts when compared to the other two harvesting periods. The iodine content of the press liquid and press cake from July was then significantly higher when compared to October, even though the content in the fresh seaweed samples was higher in October than in July, indicating that the extracted iodine content may not solely be related to the initial iodine content of the raw material and might be influenced by other factors too, such as availability and extractability. The press cake contained 30.4–58.5% lower amounts of iodine than the fresh seaweed samples, indicating that a significant amount of the iodine would be lost with the production of the press liquid. The iodine content of the seaweed meal was significantly lower in June (670 ± 113 mg/kg sample) when compared to the other two harvesting periods The high iodine content in the press liquid might also affect the possibility of usage of the chemical in food production, especially if it is dried or condensed. The recommended dietary allowance (RDA) of iodine, set by the Nordic council of ministers, for adults is 150 µg/day, and the upper-level intake is set at 600 µg/day [55]. Hence, very small amounts of fresh *A. nodosum* are needed to be consumed to obtain the RDA of iodine.

### 2.4. Total Phenolic Content (TPC) and Antioxidant Activities of Press Liquid and Seaweed Extracts

#### 2.4.1. Total Phenolic Content (TPC)

The total phenolic content (TPC) was evaluated of the freeze-dried (FD) extracts (press liquids, fresh seaweed extracts, and seaweed meal extracts) from different harvesting periods. The meal extracts from October presented the highest TPC (17.4 ± 0.7 g PGE/100 g extract), followed by the meal extracts from July and the press liquid extracts from July and October (Table 4). A significant difference was observed in the TPC between harvesting periods in both the press liquid and meal extracts, where the extracts from June contained significantly lower amounts of polyphenols. The press liquid extracts did, however, not differentiate significantly between July and October. However, the meal extracts from October contained significantly higher amounts when compared to the extracts from July. The seasonal variation in polyphenol content of *A. nodosum* has been evaluated by Apostolidis, et al. [56] and Parys, et al. [57]. In both studies, two peaks in polyphenol content were observed, one in summer (July in both cases) and another in autumn (September and October). The results from the present study also show a similar trend, or higher content detected in the extracts from July and October when compared to June, where the algae is still in its fertile period. Results have shown that there are many factors that can affect the levels of phlorotannins available in the algae, including factors such as the reproductive state of the algae, the age of the thallus, temperature, salinity of the sea, ambient nutrients, and light intensity [58,59,60]. Some studies have also indicated that herbivore grazing can increase the levels of phlorotannins in *A. nodosum* [61,62], and the grazing of seaweed in Scandinavia and Scotland seems to be at its highest in the summer [57], which can be one of the contributors to the high content of polyphenols in July. Therefore, the difference in the extracted TPC from *A. nodosum* in the present study can be linked partially to the reproductive state and seasonal variation, but other factors can also be contributors to the levels accumulated in the algae. When extraction methods were compared, a significant difference was observed in the TPC between the extracts in all harvesting periods. However, the extracts from June did not show the same results as the other two harvesting periods or indicate higher TPC when the fresh seaweed extraction was used. The other two harvesting periods showed similar results when the extraction methods were compared, where the results indicate that a higher polyphenol content was extractable by using pressure in the extraction process of the fresh seaweed. Furthermore, results from October indicated higher polyphenols were extracted from the seaweed meal and press liquid, which indicates that the potential of the alternative processing methods and the utilization of the press liquid increases as the seaweed matures during the season.

#### 2.4.2. Oxygen Radical Scavenging Activity (ORAC)

Oxygen radical absorbance capacity (ORAC) is used to determine the antioxidant capacity of an extraction by evaluating their abilities to inhibit lipid oxidation by scavenging peroxyl radicals. Thus, a higher ORAC value (Ø) indicates a stronger peroxyl radical scavenging ability. The method is based on the antioxidant abilities to inhibit oxidation induced by peroxyl radicals, which are initiated by the thermal decomposition of 2,2′-azobis (2-amidino propane) dihydrochloride (AAPH). With time, fewer antioxidants can donate hydrogen atoms to the peroxyl radicals as the reaction progresses, leading to the combining of radicals and fluorescent molecules, and hence the loss of fluorescence [63,64,65]. The results are shown in Table 5.

The highest ORAC value was observed in the fresh seaweed extract from July (1648 ± 54 μmol of TE/g extract), followed by the meal extract from July and October (1640 ± 72 and 1637 ± 179 μmol of TE/g extract). The ORAC values were significantly differentiated between harvesting periods, where June was significantly lower in ORAC values when compared to the other periods, but no difference was observed between July and October. As previously discussed, the algae were still in their fertile period in June, which led to less content of polyphenols in the extracts. ORAC values and polyphenol content have been reported to be connected with each other [14], which might explain the low ORAC activities in extracts from June. Furthermore, the same study investigated the antioxidant activities of Icelandic seaweed and showed that the Fucus species showed good antioxidant abilities, including high ORAC values. The ORAC values of water extracts of dried seaweed from *A. nodosum* harvested in May was approximately 1350 μmol of TE/g extract. The ORAC values reported by the discussed study were much higher when compared to the meal extract from June, and lower when compared to the meal extracts from July and October in the present study. The difference in values could mainly be explained by the reproductive state of the algae (which varies between April and June) [66] when compared to the values from July and October, or the method used for measurements and extraction. In the extraction of fresh seaweed extracts, slightly higher amounts of seaweed, 1.1/1 (*w*/*w*) of seaweed/water, were used compared to the amounts used in the extraction of press liquids. However, when fresh seaweed extracts are compared to press liquid, the press liquids had higher ORAC values in June and October but lower in July, indicating that the higher amounts used in the fresh seaweed extracts were not a dominating factor for the results of the ORAC measurements. In general, the results indicate that *A. nodosum* abilities to scavenge peroxyl radicals were lower in June when the seaweed is in its fertile period, compared to the other assessed seasons.

#### 2.4.3. DPPH (2,2-Diphenyl-2-picrylhydrazyl hydrate) Radical Scavenging Activity

Due to its low cost and simplicity, DPPH radical scavenging activity is one of the most popular antioxidant assays to evaluate the capacity of antioxidants to scavenge free radicals [67]. DPPH activity was evaluated for extracts from each assessed processing step obtained at different seasons at the concentrations of 5 mg/mL. The results from the DPPH assay of the seaweed samples are expressed as %inhibition (Table 5).

The DPPH assay performed on the extracts in the present study showed very high radical scavenging activity of all extracts. No significant difference was observed in the DPPH radical scavenging activities when harvesting time (June, July, and October) and the extracts were compared. The DPPH activities were similar in all extracts, ranging from 94.1% to 95.1% inhibition. Some studies have examined the DPPH radical scavenging activity of seaweed extract and brown seaweed species. Fucus species typically have rather high DPPH radical scavenging abilities compared to red and green algae [14,68]. It is difficult to compare the results of a DPPH assay to results from other studies due to the lack of a standard protocol for the assay. The results in similar studies often have different extraction methods, use other solvents during the extraction, and/or results are expressed in different units, e.g., as IC_50_ number, antiradical power (1/EC_50_), or as mg GAE/g DW. DPPH is often expressed with an IC_50_ number, which describes the concentration of an antioxidant that is required to reduce the DPPH absorbance by 50% [67]. The results in this study did not allow the calculations of an IC_50_ number since the inhibition concentration was above 75% for all concentrations measured and samples. Therefore, the DHHP radical scavenging activity results were expressed as %inhibition. The %inhibition indicates how well the extract scavenges free radicals, hence, the higher the inhibition, the better radical scavenging abilities the extract has. Free radical scavengers used in foods e.g., butylated hydroxytoluene (BHT), tocopherols, and plant phenolics have high radical scavenging activities [16,69]. Since the extracts in the present study had very high %inhabitation of DPPH, it might indicate that it could be of good use to inhibit the lipid oxidation of some food products.

#### 2.4.4. Metal Chelating Abilities (MC)

The metal chelating (MC) properties of the seaweed extracts were determined by measuring the formation of a ferrozine and Fe^2+^ complex. The metal chelating was measured at a concentration of 5 mg/mL and results are expressed as %inhibition in Table 5. The highest MC inhibition of seaweed samples was found in the seaweed meal extracts from October (78 ± 2 %inhibition), followed by the seaweed meal extracts from July (76.6 ± 2 %inhibition) and June (70.5 ± 3 %inhibition). The fresh seaweed extracts typically exhibited similar inhibition as press liquids, indicating that even though the fresh seaweed extract was made with a slightly higher seaweed content, it did not have significantly higher MC activities when compared to press liquid extracts and did not affect the results greatly. A significant difference was observed in metal chelating inhibition between extract types where the meal extracts exhibited higher MC activities when compared to the other two extraction methods. No significant difference was observed in the metal chelating abilities between harvesting periods.

The metal chelating activities of Icelandic seaweed reported by Wang, Jónsdóttir and Ólafsdóttir [14] showed that the water extracts of *A. nodosum* (in concentration of 5 mg/mL) exhibited metal chelating inhibition of approximately 95%. The study also indicated that other compounds than phenolic compounds exhibit metal chelating abilities. Other studies, such as Saiga et al. [29] and Wang, et al. [70], have reported some peptides and proteins to have metal chelating abilities. The press liquids from July and October contained some protein content, 0.3 and 0.2 g/100 g sample, respectively (Table 1), which might contribute to the metal chelating activities.

#### 2.4.5. Correlation between Total Phenolic Content and Antioxidant Activities

Principal component analysis (PCA) was performed to assess the relationship between TPC and antioxidant measurements. A PCA loading plot is shown in Figure 1 The first two components explained 48.9% and 34.4% of the total data variance, respectively. Component 1 showed a correlation between TPC and ORAC values. A Pearson correlation was performed, and the test showed a correlation between TPC and ORAC with a Pearson correlation coefficient (r) = 0.81.However, TPC did not correlate to any of the other antioxidant measurements. A study conducted by Wang, Jónsdóttir and Ólafsdóttir [14] indicated that the higher the phenolic content in a seaweed extract, the higher antioxidant capacity of the extract, including both DPPH and ORAC values. Other studies on both seaweed and plant extracts have also indicated that TPC correlates with antioxidant activities [71,72,73,74]. However, in the present study, ORAC only correlated with TPC and not to DPPH. However, Wang, Jónsdóttir and Ólafsdóttir [14] also demonstrated that other compounds, such as sterols or fucoxanthin found in brown seaweed, could also work as radical scavengers during measurements of the DPPH assay. The high DPPH antioxidant activities of the extracts in the present study may therefore be due to compounds other than polyphenols.

## 3. Materials and Methods

### 3.1. Seaweed Sampling and Preparation

*Ascophyllum nodosum* was mechanically harvested by Thorverk hf and collected in June, July, and October 2020. Approximately 20 kg of fresh seaweed from each harvest were stone cleaned by hand and minced with a Mainca CR-40 mincer (Mainca, Barcelona, Spain). Approximately 5 kg of the minced seaweed was collected in a bin, and the rest used for production of press liquid and press cake. All samples were stored at −25 °C till further use and analysis. Seaweed meal from Thorverk hf was used to examine typical chemical composition of the algal meal. The meal was processed (the biomass is harvested mechanically, transported to the factory, chopped, dried, grinded, sieved, and packed) by Thorverk from the same harvesting times as the collected fresh biomass used for the press liquid and press cake assessment. The meal was collected right after drying and stored at room temperature till further use.

### 3.2. Preperation of Press Liquid and Water Extracts

To produce the press liquid, an equal amount of tap water and fresh minced seaweed (1/1 *w*/*w*) were mixed and put through a Stephan micro cutter (Stephan Machinery GmbH, Germany). The blend was centrifuged at 5100 revolutions per minute (RPM) at 4 °C for 10 min, and both the supernatant (press liquid) and press cake were collected. The extraction of fresh minced seaweed and seaweed meal was performed with distilled water as solvent. For each seaweed meal extraction, 30 g of seaweed meal was put in a 500 mL Erlenmeyer flask, which was filled up with water to the 300 mL mark. For each fresh seaweed extraction, 150 mL of fresh seaweed and 150 mL of water (approximately 1.1/1 *w*/*w*) were put in 500 mL Erlenmeyer flask. All extracts were shaken for 1 h and centrifuged at 5100 RPM and 4 °C for 10 min. The extracts and press liquid samples were finally freeze-dried in a Genesis 25 SQ EL freeze dryer (SP industry, Philadelphia United States of America) and stored at −25 °C till further use to minimize degradation of the extract.

### 3.3. Nutritional and Mineral Composition of Seaweed and Its Produce

The water content of the seaweed samples was determined as the difference in weight of the sample before and after drying for at least 4 h at 103 ± 2 °C [75]. The total lipid content was determined according to AOCS official method Ba-3-38 (2009) [76]. The protein content of the seaweed samples was determined by using the Kjeldahl method according to ISO 5983-1:2005 [77] and ISO 5983-2:2009 [78], where the nitrogen content was multiplied by factor 5 as recommended for seaweed [27] to obtain the crude protein content in the samples. The ash content was determined by burning 2 g samples at 550 °C for 3 h, and the residue weighed and compared to the sample weight before burning [79]. The salt content was determined with the Volhard titration method [80]. Carbohydrate content of the seaweed samples was determined by calculation, subtracting water, protein, fat, and ash from 100 g of the sample.

The trace elements, arsenic, cadmium, mercury, and lead were determined according to the NMKL-186 method [81]. The inorganic arsenic content was determined by hydride generation atomic absorption spectroscopy according to the ASU method (2008-12) with modification, CON-PV 01288 (2020-05). The iodine content was determined by inductively coupled plasma mass spectrometry according to the DIN EN 15111 (2007-06) method with modification., CON-PV 01187 (2017-08).

### 3.4. Monosaccharide and Uronic Acid Composition

The monosaccharide and uronic acid contents were determined in duplicates of 25 ± 2.5 mg of the freeze-dried samples. The samples were hydrolyzed according to Wychen and Laurens [82] with sulfuric acid where 250 uL of 72% (*w*/*w*) sulfuric acid was added, the tubes were placed into 30 °C water-bath for an hour, and finally vortexed every 5 to 10 min. Afterwards, 7 mL of deionized water was added to the tubes to bring the sulfuric acid concentration to 4% (*w*/*w*). The samples were vortexed, autoclaved for one hour at 121 °C and cooled down. The hydrolyzed samples were neutralized by transferring 1 mL of hydrolysate to a white Sarstedt tube. The aliquot was neutralized to a pH 6–8 by using 1 M calcium carbonate. The aliquot was centrifuged and filtered through a 0.2 µm nylon filter (Phenomenex-Phenex, 15 mm, Torrance, CA, USA).

The monosaccharide composition of the hydrolyzed samples was analyzed according to method by Wychen and Laurens [82] and the uronic acid according to method by Basumallick and Rohrer [83]. Both measurements were performed with Dionex HPAED-PAD 5000+ on Dionex Carbo Pack PA 20 (3 × 150 mm), 6 µm column by using either two or three eluents, depending on the method used. For the monosaccharide measurements, two eluents were used, or 200 mM NaOH and deionized water, but for the uronic acid measurements three eluents were used, the same as used for the monosaccharides as well as 1 M sodium acetate in 200 mM NaOH. The flow rate of the eluents for the monosaccharide composition was set at 0.5 mL/min, and at 0.4 mL/min for the uronic acid assessment. The %monomeric sugar was calculated by using linear regression coefficients and oven dry weight of sample.

### 3.5. Total Phenolic Content (TPC)

The total polyphenol content (TPC) of the seaweed extracts were determined according to the Folin–Ciocalteu procedure as described by Singleton and Rossi [84] with slight modifications. For the measurements, 20 µL of each sample/standard (gallic acid and phloroglucinol) was put in a microplate with 100 µL of 0.2 N Folin–Ciocalteu and let sit at a room temperature for 5 min. Approximately 80 µL of 7.5% (*w*/*w*) Na_2_CO_3_ solution was then added, and the microplate put in a microwave for 10 s at 800 W. The microplate was shaken for 30 min, and then the absorbance was read at 720 nm with a Cytaion5 (Agilent Technologies, Santa Clara, CA, USA) microplate reader. The TPC was determined from the standard curves of phloroglucinol made with solutions ranging from 0 µg/mL to 100 µg/mL. Each extract was measured in triplicate (*n* = 3).

### 3.6. Antioxidant Activity

#### 3.6.1. DPPH (2,2-Diphenyl-1-picrylhydrazy) Radical Scavenging

2,2-Diphenyl-1-picrylhydrazy (DPPH) radical scavenging activity was performed according to Sharma and Bhat [85]. Quantities of 150 μL sample/ 70% (*w*/*w*) ethanol were mixed with 50 μL of 2,2 diphenyl-1-picrylhydrazy (DPPH) or 70% (*w*/*w*) ethanol in a 96-well plate. A sample was prepared by mixing the sample and DPPH solution, The microplate was covered and shaken for 30 min at room temperature at 320 rpm. The sample absorbances were read at 520 nm in a Cytaion5 (Agilent Technologies, Santa Clara, CA, USA) microplate reader. Each sample was measured in triplicate. The inhibition percentage was calculated as follows
$$\%\;inhibition=\frac{A_{blank}-\left(A_{sample}-A_{control}\right)}{A_{blank}}\times 100$$
where *A_blank_* is the absorbance of the blank, *A_sample_* is the absorbance of the sample, and *A_control_* is the absorbance of the control samples at 520 nm.

#### 3.6.2. Metal Chelating Ability (MC)

The metal chelating ability was determined according to the method described by Boyer, et al. [86] with slight modifications. A quantity of 100 µL of sample/water, 100 µL of 0.5 mM ferrozine/water, and 50 µL of 0.2 mM FeCl_2_ was added to a 96-well microplate. The samples were prepared with the sample, ferrozine, and FeCl_2_, the control sample with sample, water, and FeCl_2_, the control blank with 200 µL of water and FeCl_2,_ and the blank with water, ferrozine, and FeCl_2_. The microplate was covered and shaken for 30 min at room temperature. The absorbance was read at 560 nm with a Cytaion5 (Agilent Technologies, Santa Clara, CA, USA) microplate reader. Each sample was measured in triplicate. %Chelating activity was calculated as follows
$$\mathrm{Chelating}\;\mathrm{activity}\;(\%)=\frac{\left(A_{Net\;blank}\right)-\left(A_{Net\;sample}\right)}{\left(A_{Net\;blank}\right)}\times 100$$
where *A_Net blank_* is the absorbance of the blank minus the blank control, and *A_Net sample_* is the absorbance difference between the sample and the sample control.

#### 3.6.3. Oxygen Radical Absorbance Capacity (ORAC)

The oxygen radical absorbance capacity (ORAC assay) was analyzed according to Huang, Ou, M., Flanagan and Prior [64] and Ganske [65] with slight modifications. Approximately 60 µL of 10 mM fluorescein was added into a black 96-well microplate (Costar, cat no. 3694) along with 10 µL of sample (seaweed extract), standard (Trolox) or water (blank). Approximately 40 µL of phosphate buffer solution (pH 7.4) was added in one well for gain adjustment. The mixture was incubated at 37 °C for 10 min. After incubation, 30 µL of 120 mM 2,2 azobis (2-methylpropionamidine) dihydrochloride solution (AAPH solution) was added and the fluorescence emission (excitation at 485 nm, emission at 520 nm) was read every minute for 100 min in a POLARstar optima fluorescence analyzer (BMG Labtech, Ortenberg, Germany). The area under the fluorescence curve (AUC) was calculated by the normalized curves. Each sample was measured in triplicate.

### 3.7. Statistical Analysis

Statistical analysis of the chemical composition and antioxidant measurements was performed by using Microsoft Excel 2013 (Microsoft Inc., Redmond, WA, USA) and JMP pro15 (SAS Institute Inc., Cary, NC, USA). Averages and standard deviation (SD) were calculated in Microsoft Excel. One way analysis of variance (ANOVA) and Tukey’s honest significant difference test were performed in the JMP pro15 software (SAS, Cary, NC, USA) for each sample of each group. For all samples, statistical significance was set to *p* ˂ 0.05. Principal components analysis (PCA) was performed in JMP pro15 to evaluate the relationship between TPC and antioxidant activity of the extracts where a correlation probability test (Pearson product-moment correlation) was used to determine the correlation between variables.

## 4. Conclusions

The results from the present study indicate that the chemical composition of *A. nodosum* changes with the reproductive state of the algae and thus with harvesting seasons. Both the chemical composition and antioxidant activities of the press liquids from July and October were very similar, meaning that if the press liquid would be utilized for product development on an industrial scale, it would probably be a relatively stable production from July to October. Furthermore, due to the removal of compounds from the seaweed biomass with the production of the press liquid, the production increases the total alginate content and purity of the press cake, meaning that alginate producers could obtain an even higher yield of alginate from the pressed seaweed.

Alternative processing is more promising for high value product development from seaweed harvested in July and October compared to June due to higher total phenolic content and antioxidant activities (assessed by ORAC) in the seaweed biomass. Furthermore, product development from the ingredients present in the press liquid is in line with the current demands of full utilization and no-waste policies. The proposed alternative processing can both increase the value and sustainability of seaweed processing. Further research for applications of the press liquid is, however, needed to assess its potential for food or cosmetics applications.

## Figures and Tables

**Figure 1 marinedrugs-20-00340-f001:**
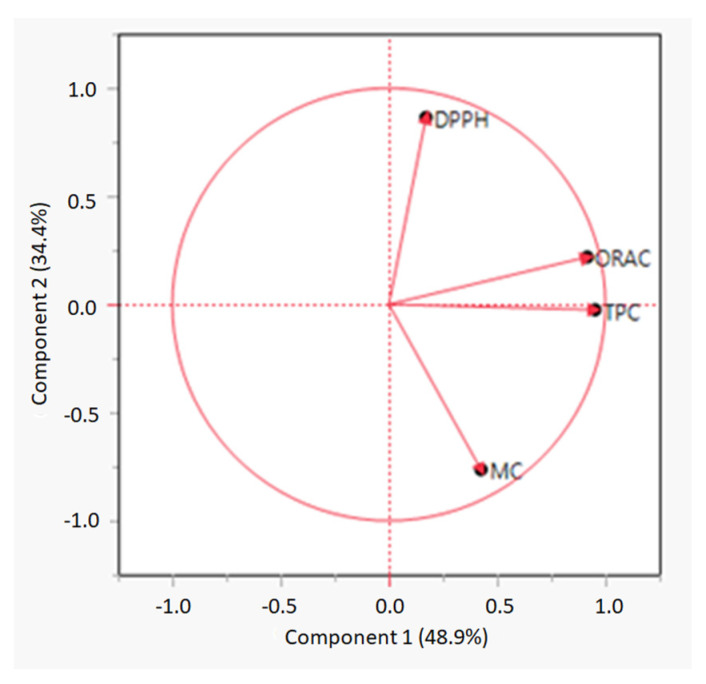
PCA loading plot of total phenolic content (TPC), oxygen radical scavenging capacity (ORAC), DPPH radical scavenging activity (DPPH), and metal chelating (MC) measurements.

**Table 1 marinedrugs-20-00340-t001:** Chemical composition of the four processing stages of seaweed during seaweed meal production. The results are expressed as mean ± standard deviation.

Harvesting Time	Sample ID	Moisture (g/100 g Sample) (*n* = 3)	Lipids (g/100 g Sample) (*n* = 4)	Protein (g/100 g Sample) (*n* = 4)	Salt (g/100 g Sample) (*n* = 4)	Salt free ash (g/100 g Sample) (*n* = 4)	Carbohydrates (g/100 g Sample) (*n* = 3)
June	Fresh seaweed	79.6 ± 0.2 ^a^	0.3 ± 0.2	1.6 ± 0.1	1.4 ± 0.0 ^b^	3.3 ± 0.1 ^c^	13.9 ± 0.2 ^c^
	Press liquid	NA	NA	NA	NA	NA	NA
	Press cake	NA	NA	NA	NA	NA	NA
	Seaweed meal	9.1 ± 0.2 ^a^	0.5 ± 0.1 ^c^	6.0 ± 0.3 ^a^	5.8 ± 0.3 ^b^	13.7 ± 0.3 ^b^	64.0 ± 2.1 ^b^
July	Fresh seaweed	70.3 ± 0.2 ^c,y^	0.2 ± 0.1	1.2 ± 0.2	1.3 ± 0.0 ^b^	4.5 ± 0.1 ^a,x^	22.5 ± 0.2 ^a,x^
	Press liquid	94.9 ± 0.1	<0.01	0.2 ± 0.0	0.9 ± 0.0	0.3 ± 0.2 ^a^	3.5 ± 0.1 ^b^
	Press cake	80.7 ± 0.1 ^x^	0.2 ± 0.1 ^b^	1.0 ± 0.1	0.6 ± 0.0	2.9 ± 0.2 ^y^	14.6 ± 0.4 ^y^
	Seaweed meal	6.9 ± 0.2 ^b^	1.6 ± 0.0 ^a^	4.7 ± 0.2 ^c^	5.8 ± 0.6 ^b^	14.1 ± 0.7 ^b^	67.0 ± 1.4 ^a^
October	Fresh seaweed	71.9 ± 0.3 ^b,y^	0.2 ± 0.2	1.1 ± 0.3	1.7 ± 0.0 ^a^	4.0 ± 0.1 ^b,x^	21.1 ± 0.4 ^b,x^
	Press liquid	94.8 ± 0.0	<0.01	0.2 ± 0.0	1.0 ± 0.0	0.2 ± 0.1 ^b^	3.8 ± 0.1 ^a^
	Press cake	80.4 ± 0.2 ^x^	0.4 ± 0.1 ^a^	0.9 ± 0.2	0.6 ± 0.0	2.9 ± 0.2 ^y^	14.9 ± 0.3 ^y^
	Seaweed meal	6.2 ± 0.2 ^c^	1.1 ± 0.1 ^b^	5.5 ± 0.2 ^b^	6.8 ± 0.1 ^a^	15.6 ± 0.2 ^a^	64.9 ± 0.2 ^b^

Different subscript letters (a–c) indicate significant differences (*p* ˂ 0.05) at each processing step between harvesting periods (June, July, October). Different subscript letters (x and y) indicate significant difference between fresh seaweed and fresh press cake within the same harvesting period (*p* ˂ 0.05). NA: Composition analyses of the press liquid and press cake from seaweed harvested in June were not performed due to the low antioxidant potential of the press liquid during this season.

**Table 2 marinedrugs-20-00340-t002:** Monosaccharide and uronic acid composition in percentage (%) of sample dry weight (dw) of the four phases of production.

		Monosaccharides				Uronic Acids	
Harvesting Time	Sample ID	Mannitol	Fucose	Glucose	Xylose/Mannose	Hexa-Mannuronic Acid	Glucuronic Acid
June	Fresh seaweed	7.1 ± 0.3 ^b^	7.0 ± 0.7 ^b^	5.1 ± 0.2 ^b^	2.8 ± 0.3 ^b^	5.1 ± 0.3	6.0 ± 0.4 ^b^
	Press liquid	NA	NA	NA	NA	NA	NA
	Press cake	NA	NA	NA	NA	NA	NA
	Seaweed meal	13.6 ± 0.9 ^b^	11.8 ± 0.9 ^a^	5.4 ± 0.5	5.1 ± 0.5	7.6 ± 0.8	8.0 ± 0.9
July	Fresh seaweed	10.4 ± 1.2 ^a^	7.4 ± 0.0 ^b^	7.1 ± 0.1 ^b^	3.5 ± 0.1 ^b^	7.1 ± 3.1	12.9 ± 0.3 ^a^
	Press liquid	41.9 ± 3.4 ^b^	1.7 ± 0.1 ^a^	12.3 ± 0.9 ^b^	1.1 ± 0.1 ^a^	ND	1.4 ± 0.1
	Press cake	8.5 ± 0.6	14.1 ± 1.0 ^b^	6.8 ± 0.3 ^b^	5.8 ± 0.4	10.4 ± 1.1	14.1 ± 2.0
	Seaweed meal	20.0 ± 1.6 ^a^	7.1 ± 3.0 ^b^	5.8 ± 2.2	3.7 ± 1.6	8.7 ± 0.5	11.0 ± 1.3
October	Fresh seaweed	11.1 ± 0.8 ^a^	11.6 ± 1.0 ^a^	11.0 ± 1.6 ^a^	4.6 ± 0.45 ^a^	8.4 ± 0.4	12.4 ± 0.3 ^a^
	Press liquid	52.4 ± 2.5 ^a^	1.2 ± 0.1 ^b^	32.8 ± 1.2 ^a^	0.6 ± 0.1 ^b^	ND	1.5 ± 0.1
	Press cake	8.2 ± 1.5	17.2 ± 1.6 ^a^	11.2 ± 1.7 ^a^	7.0 ± 1.0	9.9 ± 0.7	13.9 ± 0.5
	Seaweed meal	18.4 ± 2.3 ^a^	5.6 ± 0.4 ^b^	4.3 ± 0.5	2.7 ± 0.3	5.1 ± 0.2	6.9 ± 0.1

Different subscript letters (a,b) of same produce between different harvesting periods (e.g., mannitol of fresh seaweed from June, July, and October, and so on) indicate significant differences (*p* ˂ 0.05). NA: Composition analyses of the press liquid and press cake from seaweed harvested in June were not performed due to the low antioxidant potential of the press liquid during this season. ND: not detected

**Table 3 marinedrugs-20-00340-t003:** Arsenic, cadmium, mercury, lead, and iodine content of A. nodosum samples and their produce. The reported values can have margin of error of +/− 20% due to the method used. Results are expressed as mean ± standard deviation of wet weight (*n* = 3).

Harvesting Time	Sample ID	Mercury (mg/kg Sample)	Cadmium (mg/kg Sample)	Arsenic (mg/kg Sample)	Inorganic Arsenic (mg/kg Sample)	Lead (mg/kg Sample)	Iodine (ug/g Sample)
June	Fresh seaweed	<0.01	0.263 ± 0.006 ^a^	7.1 ± 0.2	<0.01	<0.01	156.7 ± 5.8 ^b^
	Press liquid	NA	NA	NA	NA	NA	NA
	Press cake	NA	NA	NA	NA	NA	NA
	Seaweed meal	0.023 ± 0.001	0.950 ± 0.034 ^b^	30.8 ± 0.8 ^a^	<0.01	0.039 ± 0.005 ^b^	670.0 ± 112.7 ^b^
July	Fresh seaweed	<0.01	0.28 ± 0.02 ^a^	7.3 ± 0.5	<0.01	<0.01	263.3 ± 55.1 ^a^
	Press liquid	<0.01	0.04 ± 0.0 ^a^	3.8 ± 0.01 ^a^	<0.01	<0.01	169.3 ± 1.7 ^a^
	Press cake	0.015 ± 0.001	0.18 ± 0.01 ^a^	4.4 ± 0.1 ^a^	<0.01	0.010 ± 0.005 ^b^	183.3 ± 15.8 ^a^
	Seaweed meal	0.022 ± 0.002	0.933 ± 0.026 ^b^	28.5 ± 0.5 ^b^	0.19 ± 0.01	0.056 ± 0.004 ^a^	893.3 ± 21.2 ^a^
October	Fresh seaweed	<0.01	0.083 ± 0.002 ^b^	7.5 ± 0.2	<0.01	<0.01	313.3 ± 5.8 ^a^
	Press liquid	<0.01	<0.01 ^b^	2.3 ± 0.1 ^b^	<0.01	<0.01	142.2 ± 4.1 ^b^
	Press cake	0.014 ± 0.002	0.063 ± 0.004 ^b^	3.7 ± 0.1 ^b^	<0.01	0.023 ± 0.002 ^a^	130.0 ± 0.0 ^b^
	Seaweed meal	0.020 ± 0.002	1.43 ± 0.03 ^a^*	31.3 ± 1.0 ^a^	0.12 ± 0.00	0.035 ± 0.002 ^b^	980.0 ± 20.0 ^a^

Different subscript letters of same produce between different harvesting periods (e.g., mercury of fresh seaweed from June, July, and October, and so on) indicate significant differences (*p* ˂ 0.05). An asterisk (*) indicates values above the set limits according to European Commission for dried animal feed. NA: Composition analyses of the press liquid and press cake from seaweed harvested in June were not performed due to the low antioxidant potential of the press liquid during this season.

**Table 4 marinedrugs-20-00340-t004:** Total phenolic content (TPC) of the freeze-dried extracts (fresh seaweed extracts, press liquid extracts and seaweed meal extracts as affected by harvesting season. Results are expressed as mean ± standard deviation.

Harvesting Time	Extracts	TPC * (g PGE/100 g Extract) (*n* = 3)
June	Fresh seaweed extract	12.0 ± 0.2 ^a^
	Press liquid extract	10.8 ± 0.5 ^b,y^
	Seaweed meal extract	11.0 ± 0.2 ^b,y^
July	Fresh seaweed extract	13.1 ± 0.3 ^b^
	Press liquid extract	15.2 ± 0.7 ^a,x^
	Seaweed meal extract	15.5 ± 0.8 ^a,y^
October	Fresh seaweed extract	12.5 ± 0.8 ^c^
	Press liquid extract	15.1 ± 0.1 ^b,x^
	Seaweed meal extract	17.4 ± 0.7 ^a,x^

Different subscript letters (a–c) within columns indicate significant differences in extraction method (*p* ˂ 0.05). Different subscript letters (x,y) of same produce between different harvesting periods (e.g., FSE/PL/SME from June, July, and October) indicate significant differences (*p* ˂ 0.05). * A typical equation for the standard curve was 3.7741x + 0.0316, with the correlation coefficient R^2^ = 0.9983, a level of detection LOD = 0.010026, and level of quantitation LOQ = 0.432747.

**Table 5 marinedrugs-20-00340-t005:** Oxygen radical scavenging activity (ORAC), DPPH radical scavenging activity (DPPH), and metal chelating ability (MC) of extracts from different harvesting periods. Results are expressed as mean ± standard deviation (*n* = 3). Measurements of DPPH and MC were made with concentration of 5 mg/mL.

Harvesting Time	Extracts	ORAC Value (μmol of TE/g Extract)	DPPH (% Inhibition)	MC (% Inhibition)
June	Fresh seaweed extract	507 ± 11	94.1 ± 0.3	50.9 ± 1.3
	Press liquid extract	625 ± 9	94.8 ± 0.3	50.2 ± 3.8
	Seaweed meal extract	680 ± 71	94.8 ± 0.4	70.5 ± 2.9
July	Fresh seaweed	1648 ± 54	95.1 ± 0.5	49.2 ± 2.8
	Press liquid extract	1452 ± 62	94.9 ± 0.4	44.2 ± 4.9
	Seaweed meal extract	1640 ± 72	93.4 ± 0.3	76.6 ± 1.9
October	Fresh seaweed extract	1314 ± 143	94.6 ± 0.1	44.7 ± 3.8
	Press liquid extract	1476 ± 109	95.3 ± 0.5	45.7 ± 3.7
	Seaweed meal extract	1637 ± 179	94.9 ± 0.2	78.0 ± 1.8

## Data Availability

Not applicable.

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
