# Peer review of "Added Value of Ascophyllum nodosum Side Stream Utilization during Seaweed Meal Processing"

_marinedrugs, 2022, doi:10.3390/md20060340_

Round 1

Reviewer 1 Report

the manuscript "Added value of Ascophyllum nodosum side stream utilization 
during seaweed meal processing" is well organized, strutured and very interesting. There is only some notes in the annexed document for the authors revise, mostly in introduction.

Author Response

We would sincerely like to thank the reviewers for their constructive comments on our manuscript. All changes in the manuscript have been marked with blue, and all lines mentioned in the responses refer to the line numbering in the revised manuscript.

Point 1: Add FAO values of worlds seaweed production from 2018.

Response 1: The following sentence has been added about seaweed production from 2018 to introduction, as suggested by the reviewer: The production continues to increase and in the year 2018 the worlds seaweed production were 32.4 million tons, where 97.1% were obtained from farmed seaweeds. (lines 45-47)

Point 2: Food industry, the agar, carrageenan and alginate extraction industry uses wild biomass, please revise. the same for agriculture and animal feed, one example is Ascophyllum nodosum which is not "cultivated" but harvested in the wild although the harvest method permits the seaweed easy regeneration.

Response 2: The following sentence has been taken out of the manuscript to support the comment of the reviewer: “Therefore, wild harvested seaweed is currently only a very small part of the industry [1].” (line 47).

Point 3: Examples of bioactive compounds in text

Response 3: The following examples of bioactive compounds have been added to the sentence: (including polyphenols, peptides, and carotenoids) (line 50-51)

Point 4: some – delete

Response 4: The word “some” has been deleted from the text, as requested. (line 58)

Point 5: sulphated polysaccharides – change to low molecular weight sulphated polysaccharides.

Response 5: The term “sulphated polysaccharides” has been changed into “low molecular weight sulphated polysaccharides” in the text, as suggested. (line 59)

Point 6: revies that leftover pulp from alginate production is thrown away – it is mostly sent to landfills.

Response 6: The sentence has been changed to: “The leftover pulp from hydrocolloid production is usually sent to landfills, pumped back to the sea, or occasionally used as soil conditioner [2, 10, 11], meaning that around 70% of the dried seaweed biomass is not utilized to its full potential.” (lines 68-71)

Point 7: change alginate into sterols or cellulose, since alginate in their native form is very soluble in water.

Response 7: The term “alginate” has been changed to “cellulose” in the sentence. (line 78)

Point 8: why NA in table of chemical composition of press liquid and press cake.

Response 8: The following sentence was added to the sub-text of Tables 1 and 3 to explain the missing values: “Composition analysis of the press liquid and press cake from seaweed harvested in June were not performed due to the low antioxidant potential of the press liquid during this season” (line 105-106, and lines 212-213, respectively) The antioxidant potentials of seaweed extracts were confirmed before chemical analysis of the seaweed, leading to the decision to focus on the chemical composition of the side product in the other harvesting periods.

Point 9: there was field observation for this seaweed reproduction systems?? Or water parameters which can be observed water quality to interfere the water physico-chemical characterization? if not, be careful, with your analysis

Response 9: Yes, there were field observations at the harvest, where the seaweed collected in June were observed to contained fruit bodies. To place emphasis on the fertile period of Ascophyllum in June in the research, the following sentences (lines 112-114) were changed in the article:  “The water content might hence be related to the reproduction stage of the algae since fruit bodies/receptacles were observed on the seaweed in June, indicating that the seaweed was in its fertile period [18]. (lines 112-114)

Point 10: Reference in sentence in line 147-149.

Response 10: References have been added (line 161 in revised manuscript).

Point 11: Reference for sentence in line 272-273: Some studies investigating Sargassum species have shown changes in alginate content over the year, reaching its peak in the summer.

Response 11: Sentence was removed to shorten the results and discussion as proposed by reviewer 2. 

Point 12: Reference line 370-372, The recommended dietary allowance (RDA) of iodine set by WHO and Nordic council of ministers for adults is 150 μg/day, and upper-level intake is set at 600 μg/day.

Response 12: Reference added for the Nordic recommendations 2012, as requested. (line 387)

Point 13: reference of the ISOs - ISO [72] and ISO [73], line 564.

Response 13: Reference changed to according to ISO 5983-1:2005 [77] and ISO 5983-2:2009 [78], (line 581 in revised manuscript)

Point 14: name of authors in line 580 for monosaccharide composition method.

Response 14: Names of authors added - Wychen and Laurens [82]  (line 597-598)

Reviewer 2 Report

The aim of the study was to investigate the chemical and bioactive characteristics of A.nodosum as affected by seasonal variation and evaluate the potential benefits of alternative processing and utilization of side streams for product development.

The paper is well written and so I suggest to publish it in this form.

The only address I have is that the bibliography is not updated ando so I suggest Authors to add more papers in the range 2019-2022.

Author Response

We would sincerely like to thank the reviewer for the constructive comments on our manuscript. All changes in the manuscript have been marked with blue, and all lines mentioned in the responses refer to the line numbering in the revised manuscript.

The paper is well written and so I suggest to publish it in this form.

Point 1: The only address I have is that the bibliography is not updated ando so I suggest Authors to add more papers in the range 2019-2022.

Response: Four papers were added to bibliography from the years between 2017-2022. These are the following:

Buschmann, A. H.; Camus, C.; Infante, J.; Neori, A.; Israel, Á.; Hernández-González, M. C.; Pereda, S. V.; Gomez-Pinchetti, J. L.; Golberg, A.; Tadmor-Shalev, N.; Critchley, A. T., Seaweed production: overview of the global state of exploitation, farming and emerging research activity. European Journal of Phycology 2017, 52, (4), 391-406. [reference 3 in revised manuscript]

Mannino, G.; Campobenedetto, C.; Vigliante, I.; Contartese, V.; Gentile, C.; Bertea, C. M., The Application of a Plant Biostimulant Based on Seaweed and Yeast Extract Improved Tomato Fruit Development and Quality. 2020, 10, (12), 1662. [reference 4 in revised manuscript]

Kumar, S.; Sahoo, D., A comprehensive analysis of alginate content and biochemical composition of leftover pulp from brown seaweed Sargassum wightii. Algal Research 2017, 23, 233-239. [reference 12 in revised manuscript]

FAO, The State of World Fisheries and Aquaculture 2020. FAO: Rome, Italy, 2020. [reference 6 in revised manuscript]

Reference numbers and citations to references in the manuscript have been updated accordingly to these additions and where they appear in the manuscript.

Reviewer 3 Report

The manuscript entitled “Added value of Ascophyllum nodosum side stream utilization during seaweed meal processing”, authored by Anna Þóra Hrólfsdóttir and colleagues, deals with the investigation of the chemical and bioactive characteristics of A. nodosum as affected by seasonal variation and the evaluation of the potential benefits of alternative processing and utilization of side streams for product development.

The article contains really interesting data that can seriously contribute to the current state of the art. In addition, the manuscript is written with extreme authority, although several grammatical errors and typos are present in the main text. However, I do not feel that this will contribute to the rejecting of the manuscript, but I strongly advise the authors to revise their papers before resubmitting it to the journal.

Below is a list of observations that the authors should implement.

AFFILIATIONS: Affiliations lack the acronyms of each author after reporting the email address. The same acronyms should be used for the contributions section at the end of the manuscript.

ABSTRACT: The abstract is really well written and organized. The only problem is related to the number of words used. As clearly stated in the guidelines for authors provided by the journal, a maximum of 200 words is allowed for this section. Consequently, authors should try to reduce this section as much as possible.

KEYWORDS: regarding the keywords, they are a useful tool to help indexers and search engines to find relevant papers of interest. If scientific search engines (such as PubMed, Scopus, Google Scholar, etc) can find a potential manuscript by the use of words contained in both title, abstract, and keywords. Consequently, readers will be able to find it too thank this words. An easier search of the manuscript allows to increase the number of people reading your manuscript after publication and, then, to obtain more citations. Consequently, keywords should be words preferably not contained in the title or abstract. This short explanation is to suggest that authors introduce as many keywords as they can, and replace those words that are already present at least in the title with new keywords properly related to the reviewed manuscript.

INTRODUCTION:

  • At lines 30-35, the authors should include information regarding the use of algae as ingredients in biostimulant formulations used in agriculture, both in order to increase the nutraceutical properties of fruits for human consumption (10.3390/biom10121662; 10.18805/LR-412) and plant resilience to abiotic stresses (doi.org/10.3390/agriculture11060557; 10.1007/s13762-021-03568-9).
  • LINE 57: is harvested mechanically -> is mechanically harvested

RESULTS AND DISCUSSION: The results and discussion section is really well written. In particular, the authors clearly described and commented on the obtained results, comparing them to previously published data. My main concern about this section relates to the too lengthy, sometimes unnecessary, sometimes far too repetitive commentary of the results. The authors should therefore try to shorten this section slightly in order to make it easier to read by interested scientists.

MATERIALS AND METHODS:

  • LINE 545: (1/1 in weight) -> (1/1 w/w)
  • Subsection 3.3. should be renamed as “Nutritional and Mineral Composition of….”
  • LINE 579: percentage numbers should always be followed by the ratio specification (e.g., 72/ v/v of sulfuric acid. Please, fix the problem all over the main text.
  • LINE 609: what standard was used for calibration curve? Please, provide the equation of the curve, along with statistical data (R2, LOD, LOQ).
  • Were the results of the antioxidant assays expressed as % inhibition or in reference to a pure standard such as gallic acid or trolox? please provide this information for each of the used assays in the respective subsection.

CONCLUSION: The concluding section should be a short paragraph reporting and summarizing the main results obtained. A maximum of 10-15 lines should form this section. The authors should drastically reduce this section, and report the deleted information if it is not in the discussion section.

The paragraph of data availability is missing. Please introduce it.

Author Response

We would sincerely like to thank the reviewer for the constructive comments on our manuscript. All changes in the manuscript have been marked with blue, and all lines mentioned in the responses refer to the line numbering in the revised manuscript.

Point 1: AFFILIATIONS: Affiliations lack the acronyms of each author after reporting the email address. The same acronyms should be used for the contributions section at the end of the manuscript.

Response 1: Acronyms and emails have been added to affiliations as requested.

Point 2: ABSTRACT: The abstract is really well written and organized. The only problem is related to the number of words used. As clearly stated in the guidelines for authors provided by the journal, a maximum of 200 words is allowed for this section. Consequently, authors should try to reduce this section as much as possible.

Response 2: Minor changes were made to the abstract phrasing to fulfil the maximum word criterion.

Point 3: KEYWORDS: regarding the keywords, they are a useful tool to help indexers and search engines to find relevant papers of interest. If scientific search engines (such as PubMed, Scopus, Google Scholar, etc) can find a potential manuscript by the use of words contained in both title, abstract, and keywords. Consequently, readers will be able to find it too thank this words. An easier search of the manuscript allows to increase the number of people reading your manuscript after publication and, then, to obtain more citations. Consequently, keywords should be words preferably not contained in the title or abstract. This short explanation is to suggest that authors introduce as many keywords as they can, and replace those words that are already present at least in the title with new keywords properly related to the reviewed manuscript.

Response 3: Keywords have been changed from “Seaweed, Processing, Ascophyllum nodosum; Proximal composition; Monosaccharide composition; Bioactive compounds; Polyphenols; Antioxidant activity” to the following keywords: “Ascophyllum nodosum; Brown seaweed, Macroalgae; Seasonal variation; Proximal composition; Trace minerals; Monosaccharide composition; Bioactive compounds; Polyphenols; Antioxidant activity”

Point 4: INTRODUCTION: At lines 30-35, the authors should include information regarding the use of algae as ingredients in biostimulant formulations used in agriculture, both in order to increase the nutraceutical properties of fruits for human consumption (10.3390/biom10121662; 10.18805/LR-412) and plant resilience to abiotic stresses (doi.org/10.3390/agriculture11060557; 10.1007/s13762-021-03568-9).

Response 4: The following sentence has been added to the introduction to include the suggested references: “[1-3]. Besides direct consumption, seaweed is used as an ingredient in bio-stimulants for agriculture, to enhance nutraceutical properties of plants and fruits for human consumption [4], and to increase plant resilience to abiotic stresses [5]” (lines 38-41)

Point 5: INTRODUCTION: LINE 57: is harvested mechanically -> is mechanically harvested

Response 5: The sentence has been changed according to the reviewer’s suggestion (line 64).

Point 6: RESULTS AND DISCUSSION: The results and discussion section is really well written. In particular, the authors clearly described and commented on the obtained results, comparing them to previously published data. My main concern about this section relates to the too lengthy, sometimes unnecessary, sometimes far too repetitive commentary of the results. The authors should therefore try to shorten this section slightly in order to make it easier to read by interested scientists.

Response 6: Thank you for the comment. The results and discussion have been shortened slightly in chosen sections to avoid repetition and to give the section more focus. Deleted sentences are marked with Track-changes in the revised manuscript.

Point 7: MATERIALS AND METHODS: LINE 545: (1/1 in weight) -> (1/1 w/w)

Response 7: The sentence has been changed according to the reviewer’s comment: (1/1 in weight) changed into (1/1 w/w)

Point 8: MATERIALS AND METHODS: Subsection 3.3. should be renamed as “Nutritional and Mineral Composition of….”

Response 8: The subsection title has been changed in agreement with the reviewer’s suggestion (line 576)

Point 9: MATERIALS AND METHODS: LINE 579: percentage numbers should always be followed by the ratio specification (e.g., 72/ v/v of sulfuric acid. Please, fix the problem all over the main text.

Response 9: The ratio specifications have been updated as appropriate throughout the manuscript, as requested.

Point 10: MATERIALS AND METHODS: LINE 609: what standard was used for calibration curve? Please, provide the equation of the curve, along with statistical data (R2, LOD, LOQ).

Response 10: The description of the TPC content assessment has been changed to be more descriptive to the following: “For the measurements, 20 µL of each sample/standard (gallic acid and phloroglucinol) was put in a microplate …..” (line 621-622)

And changes to Line 627/628 changed include that the TPC was determined from the standard curves of phloroglucinol.

Furthermore, the following information on the standard curve has been added to the table text for Table 4.

“A typical equation for the standard curve was 3.7741x + 0.0316, with the correlation coefficient R2 = 0.9983, a level of detection LOD = 0.010026, and level of quantitation LOQ = 0.432747.”

Point 11: MATERIALS AND METHODS: Were the results of the antioxidant assays expressed as % inhibition or in reference to a pure standard such as gallic acid or trolox? please provide this information for each of the used assays in the respective subsection.

Response 11: Equations for calculating the %inhibitation and %chelating activity of DPPH and Metal chelating were added to text:

For DPPH:

“The inhibition percentage was calculated as follows:

% inhibition =

where Ablank is the absorbance of the blank, Asample is the absorbance of the sample and Acontrol is the absorbance of the control samples at 520 nm.” (lines 638-644)

For metal chelating the following text was added:

%Chelating activity was calculated as follows:

Chelating activity (%) =  

where ANet blank is the absorbance of the blank minus the blank control, and ANet sample is the absorbance difference between the sample and the sample control.

Point 12: CONCLUSION: The concluding section should be a short paragraph reporting and summarizing the main results obtained. A maximum of 10-15 lines should form this section. The authors should drastically reduce this section and report the deleted information if it is not in the discussion section.

Response 12: The conclusions section has been shortened and rewritten to summarize the main results and conclusions of the study, in agreement with the reviewer’s comments.  (Lines 685-701)

Point 13: The paragraph of data availability is missing. Please introduce it.

Response 13: The following sentence on data availability has been added to the manuscript:

“Data availability: Further data and information are available upon request to the corresponding author.” (lines 718-719).